# Major Common Hallmarks and Potential Epigenetic Drivers of Wound Chronicity and Recurrence: Hypothesis and Reflections

**DOI:** 10.3390/ijms26178745

**Published:** 2025-09-08

**Authors:** Alicia Tamayo-Carbón, Ariana García-Ojalvo, José Fernández-Montequín, William Savigne-Gutiérrez, Gretel de Armas-López, Cristina Carbonell-López, Sheila Montero-Alvarez, Dionne Casillas-Casanova, Gabriela Pino-Fernández, Jorge Berlanga-Acosta

**Affiliations:** 1Hospital Clínico-Quirúrgico Hermanos Ameijeiras, Calle San Lázaro No. 701 esq. a Belascoaín, Centro Habana, Havana 10200, Cuba; aliciatamayo67@gmail.com; 2Tissue Repair and Cytoprotection Research Group, Biomedical Research Direction, Center for Genetic Engineering and Biotechnology, Ave. 31 S/N. e/158 and 190, Cubanacán, Playa, Havana 11300, Cuba; ariana.garcia@cigb.edu.cu (A.G.-O.); sheila.montero@cigb.edu.cu (S.M.-A.); dionne.casillas@cigb.edu.cu (D.C.-C.); gabriela.pino@cigb.edu.cu (G.P.-F.); 3Instituto Nacional de Angiología y Cirugía Vascular, Calzada del Cerro No. 1551 esq. Domínguez, Cerro, Havana 10500, Cuba; montequi@infomed.sld.cu (J.F.-M.); william.savigne@infomed.sld.cu (W.S.-G.); 4Complejo Científico Ortopédico Internacional Frank País, Avenida 51 No. 19603, La Lisa, Havana 17100, Cuba; greteldearmaslopez@gmail.com (G.d.A.-L.); ccarbonell79@gmail.com (C.C.-L.)

**Keywords:** chronic wounds, diabetic foot ulcer, pressure injury, venous ulcer, ulcer recurrence, epigenetic, wound healing

## Abstract

Chronic wounds are considered a silent epidemic that impact millions of human lives worldwide, causing comorbidities, reducing life quality and expectancy. Diabetic, pressure, and venous ulcers are the three major clinical entities of chronic wounds, in which the presence of a chronicity phenotype and episodes of recurrence remain as contemporary challenges. We are, accordingly, far from a full understanding about the potential endogenous, predisposing factors that may drive both chronicity and recurrence. Decades of academic and financial endeavors have not translated into a pharmacological intervention that may curb these events. These wounds may exhibit the clinical aspect of a torpid granulative response, poor angiogenesis, delayed or abnormal re-epithelialization, and low contraction rates. At the cellular level, chronicity is propelled and distinguished by the triad of interplaying loops of inflammation, oxidative stress, and cellular senescence. Although the proximal molecular drivers of chronicity and their hierarchal debut sequence are a critical research target and pending task, our unifying hypothesis behind chronicity and recurrence is founded on the existence of an epigenetic pathologic code that originates and perpetuates a “chronic wound memory”. In vitro studies suggest that this *de novo* edited script is sheltered in dermal fibroblasts and keratinocytes and is spreadable and transmissible to descendant cells, dictating abnormal traits even in ideal culture conditions and successive passages. The list of epigenomic alterations and their significance in wound pathology is continuously escalating. The accurate identification of the key epigenetic priming codes of impaired healing, and their selective re-editing, will be remarkably beneficial.

## 1. Introduction

As the human body’s largest organ with an approximate area of 30 m^2^ and a first line of defense against the external environment, the skin is evolutionarily endowed with many proliferation-responsive cells and an urgent repair mechanism before injuries [1,2]. This organ is conditioned to host a peculiar microbiota and to actively manufacture and respond to a repertoire of cytokines, growth factors, chemokines, and neurotransmitters in health and disease conditions [3]. This constellation of soluble signalers plays a critical role during cutaneous wound healing in a sequence of finely coordinated events in time and space [4]. Central to the physiologic skin repair process is the heterogeneous population of dermal fibroblasts with unprecedented plasticity and memory capabilities. These are critical cells in the “response to injury” and also central is the role of the so-called “diseased fibroblasts” in torpid healing and consequently in wound chronicity [2,5,6]. The outcome of the cutaneous healing process is influenced by a myriad of external and internal factors that may ultimately drive three major products: physiologic, exuberant, and deficitary healing. Both exuberant and deficitary healing are pathologic conditions. The former is the result of an evolutionary pressure to heal through the rapid formation of fibrotic tissue [7,8,9], whereas the latter includes chronic or hard-to-heal wounds. Considered a “silent epidemic”, chronic wounds (CWs) are conceptually defined as wounds that have not reduced in size by more than 40% to 50% or healed within 1 month [10], exhibiting a slow rate of in size reduction (≤1 mm/week) [11]. CWs may be classified as having primary refractoriness when, since its early phases, there is no clinical evidence of transition, whereas secondary refractoriness entails those wounds that after having entered the proliferative phase become arrested with no apparent cause. They often present with granulation tissue but lack re-epithelialization [12]. These conditions impact the life of nearly 2.5% of the total population of the United States, and this fraction is progressively escalating in elder and diabetic populations [13,14]. These lesions are associated with anxiety and depression, both of which hinder the molecular foundations of the healing process, predispose to and prolong infection, and increase ulcer recurrence rates [15,16,17,18]. From a financial perspective, wound healing is the largest medical market without an existing small molecule/drug treatment. The US medical burden is estimated to be over USD 25 billion every year [19]. These data highlight our incomplete understanding of the molecular and cellular fundamentals of wound chronicity and, at the same time, explain the disappointing results of modern therapies [20,21].

For years, wound chronicity investigations have mostly focused on the identification and validation of external factors such as poor glycemic control for the case of diabetic populations, toxic habits, wound anatomical location, endovascular low-grade inflammation, insufficient nutrient assimilation/catabolic phenotype, neuropathy, ischemia, and mechanical pressure [22]. In an attempt to gain further insights into the biological conundrum of CWs, we reviewed the recent literature, aiming to unravel the most common intrinsic mechanisms operating behind the refractoriness in healing and the propensity to recurrence in the three major types of human CW. Accordingly, we advocate the unifying hypothesis based on the existence of a “CW memory” or “open portal memory” which has been progressively developed and may act as an endogenous driving force behind chronicity and recurrence [23]. The search strategy involved Medline/PubMed and Google Scholar data sources using the following key words: wound healing, chronic wounds, diabetic, venous, and pressure ulcers, wound/ulcer recurrence, fibroblasts senescence, and fibroblasts + proliferative arrest. Only English language articles were included.

## 2. Invariants of the Most Frequent Chronic Wounds

Based on the causative etiologies, CWs have been classified into four major categories: pressure injuries (PIs), diabetic foot ulcers (DFUs), venous leg ulcers (VLUs), and atypical hard-to-heal wounds such as those derived from radiation injury, chemotherapy extravasation, arterial and Buruli ulcers, etc. [13,24]. Although we and others have described specific histological markers for each of the most frequent forms of human CW, [11,25] in general terms, they exhibit five major clinical invariants: (1) delayed onset and growth of granulation tissue (GT), (2) persistent inflammation and propensity to infection, (3) torpid re-epithelialization [26], (4) systemic clinical repercussions [27], and (5) propensity to short-term recurrence after re-epithelialization is completed, even in those scenarios where external risk/predisposing factors are controlled [28].

## 3. Brief Depiction of the Three Main Chronic Wounds

### 3.1. Diabetic Foot Ulcers

DFUs remain as the main cause of limb amputations worldwide, accounting for 80% of all non-accidental amputations [29]. Furthermore, about 50% of diabetics who undergo amputations die within the next five years [19]. These ulcers are prone to recurrence by a combination of external and internal factors [30] so that about 40% of patients will experience recurrence within the first year of re-epithelialization, whereas this figure expands to 60% during the next three years post-healing [31]. The most common DFU classification systems which have been broadly validated for ulcer healing and lower extremity amputation occurrence include Meggitt–Wagner, SINBAD, University of Texas, and WIfI [32,33]. Impaired tissue repair is a diabetic complication and one of the clinical consequences of long-lasting poor glycemic control and the ensuing biochemical derangements, which ultimately create a systemic cytotoxic environment [34]. Converging studies show the deleterious role of reactive oxygen species (ROS)/oxidative stress (OS) as major actors in the pathogenic cascade of diabetic tissue damage [35,36]. Hyperglycemia and its metabolic-associated derivatives, such as advanced glycation end-products (AGEs), accumulate in the dermal collagen and impair fibroblast physiology, provoking precocious cutaneous aging, while perpetuating a chronic inflammatory infiltrate associated with extracellular matrix degradation by local proteases [37]. In this scenario, locally secreted growth factors and their receptors are degraded, imposing an additional restrictive factor for cellular proliferation [38,39,40]. Glucotoxicity induces cellular senescence, disturbs collagen expression, causes apoptosis of dermal fibroblasts and endothelial progenitor cells, and upregulates RELA/p65 expression which in turn amplifies proinflammatory, oxidative, and prodegradative profiles, thereby limiting GT progression (Figure 1) [41,42].

Hyperglycemia is known to act as a senescence-promoting factor for skin cells [43]. The notion that cellular senescence is an imperceptible underlying actor in the pathogenesis of wound chronicity and ulcer recurrence has been supported for years [44]. As discussed below, the establishment of a senescence cell society, especially of “diseased fibroblasts”, and the dysfunctionality of stem cell populations are significant pathophysiological ingredients for diabetic wound chronicity [45,46]. Another exciting and expanding research field focuses on the instrumental role of the diabetes-associated epigenetic landscape for the onset and irreversibility of diabetic complications, including impaired healing [47]. This hyperglycemia-derived imprinting acts as the foundation and propeller of metabolic memory, which in turn perpetuates the senescent phenotype in fibroblasts and keratinocytes through an inflammotoxic secretome [48,49,50]. Histopathological differences exist in the aspect and composition of the GT between neuroinfectious and ischemic ulcers, irrespective to the type of diabetes and the evolution data. The mechanisms behind the differences of each type of extracellular matrix, angiogenic response, organizational pattern, vascular density, and even in fibroblast cytology have not been elucidated. However, the acute and faithful recapitulation of a series of vascular anomalies which are of long-term evolution [25], such as media thickening, neointimal hypertrophy, and incomplete angiogenic organization within an early emerging GT, may be evocative of the existence and impact of the metabolic memory in the cells committed to wound repair [51,52]. Ironically, it is likely that diabetic ulcer scar tissue harbors in its core the endogenous biological drivers of recurrence [30].

### 3.2. Pressure Injury

In 2016, the United States National Pressure Ulcer Advisory Panel, now the National Pressure Injury Advisory Panel, recommended changing terminology from pressure ulcer to pressure injury, attempting to provide a better description of the stages that do not appear within the open wound [53]. This condition, also known as pressure sores, decubitus ulcers, or simply decubitus, has been known since ancient Egypt and has challenged the best medical and nursing procedures for years [54]. Today, PIs still pose enigmas and prompt interrogations about its pathophysiology [55]. Thus, thought-provoking questions remain to be answered: (1) Why do PIs not heal in a physiological manner considering that most PI patients are not diabetics? (2) Why do PIs not follow a normal healing trajectory once ischemia has been relieved? (3) What are the driving factors propelling the overwhelming cases of recurrence [56]? These facts have incited us to hypothesize on the existence of a “local, long-lasting pressure memory” which conveives that like for other CWs, and there are intrinsic, endogenous factors leading to chronicity and recurrence [57]. PIs are defined as injuries on the surface of the skin or in the underlying tissue, usually over a bone prominence, caused by prolonged pressure, friction, or shearing force [58]. They are a serious complication of multimorbidity and lack of mobility and remain as an important source of psychological distress for patients and caregivers, especially when the major populations at risk are older subjects with increased vulnerability [59,60]. A recent meta-analysis indicated the global prevalence of PIs to be 12.8%, with an in-hospital acquired incidence of 8.4% [61]. The reported prevalence in the United States reaches the alarming figure of 2.5 million [62]. Thus, PIs represent a tremendous burden to the health economy, accounting for 4% of total national health system expenditure in developed countries [63]. The developmental cascade of PI onset is a complex, progressive, and multifactorial process involving extrinsic factors such as prolonged pressure, friction, shear force, and moisture/tissue maceration. Intrinsic factors include systemic inflammation, cachexia, catabolic states, malnutrition/hypoproteinemia, anemia, hypoxia, and endothelial dysfunction. Patients with older age, cognitive impairment, and comorbid conditions and in general patients with the inability to respond to discomfort-related nervous input signals are predisposed to PIs [64,65]. These lesions are usually largely exudative, especially when accompanied by local bacterial colonization and inflammatory reaction, and their clinical severity is graded through an internationally acknowledged scale from grades 1 through 4 [66,67]. A variety of complementary etiopathogenic theories have emerged to explain PI onset [68]. However, long-enduring or intermittent soft tissue pressure beyond a tolerance threshold remains as the critical driving force for the onset of ischemic necrosis [69]. Pressure of more than 33 mmHg occludes the blood vessels, induces cell death in the skin and the underlying tissues, and eventually contributes to PI development [70]. Histopathologic findings describe a hierarchical damage cascade (Figure 2) in which papillary dermis vessels are primarily targeted by pressure/ischemia, whereas the major damage is observed in the muscle, followed by subcutaneous tissue and then skin upper layers. This sequence of damage is likely a reflection of each tissue stratum’s metabolic requirement/hypoxia tolerance [64]. Alike other CWs, PI recurrence represents a research challenge and social problem for contemporary medicine [71,72,73]. The incidence of recurrent pressure injuries is 5.4–73.6% [74]. The most commonly reported site of recurrence is the ischium [75,76]. Moreover, recurrence rates after flap surgery have reached 82% with no trend toward improvement during the last 50 years despite medical and nursing advancements [77,78]. Although the number of studies on the identification of risk/predictive factors for PI recurrence are descriptive and often conflicting [79], among the plethora of described factors defect size/damage degree and serum albumin levels deserve credibility [80,81]. Nevertheless, PI is a paradigmatic model of a chronic and recurrent wound, which throughout its evolution may shape the endogenous epigenetic drivers that may predispose to recurrence [68,82].

### 3.3. Venous Leg Ulcers

In our view, VLUs exhibit the most intertwined and variegated pathophysiological mechanism (Figure 3). The ulcers are the outward manifestation of a long-term chronic venous disease (CVD) and they are the most common CW in the lower extremity [83,84], deteriorating quality of life and imposing a high level of frailty and comorbidities [85]. Quality of life is further affected by pain, infection, inflammation, and odor [86]. Common to other CWs, these distressing situations generate depression, anxiety, and sleeping disorders which negatively impact on a patient’s health [87].

The Clinical–Etiological–Anatomical–Pathophysiological (CEAP) classification is the current international standard for classifying CVD. The cornerstone of treatment for venous ulcers is compression therapy [88]. Behind the debut of a VLU lies a very complex and intricate pathophysiological mechanism, involving a cascade of hemodynamic, cellular, and molecular alterations of macro- and microcirculation, which are ultimately transmitted to the skin. Proximal to this complex scenario, the major driver is the presence of venous insufficiency leading to venous hypervolemia/hypertension/reflux [89]. High pressure of deoxygenated venous blood causes “venous ischemia and hypoxia”. The hypertension leads to fibrinogens and extravasation of a plethora of biomolecules to perivenous space, provoking dermal edema and the formation of fibrin cuffs, which further amplify hypoxia and vein wall damage [90]. Hypertension/reflux alters shear stress, resulting in endothelial damage due to the overexpression of adhesion molecules (interstitial cell adhesion molecule 1 (ICAM-1), vascular cell adhesion molecule 1 (VCAM-1), and E-selectins), the release of inflammatory mediators and metalloproteases, endovascular cell recruitment, and activation of prothrombotic precursors [28]. Venous hypertension is responsible for endothelial-to-mesenchymal transition via activation of BMP4/pSMAD5-SNAI1/2 signaling [91] which together with target fibroblasts, vascular smooth muscle cells, and the extracellular matrix creates a local fibrosing and sclerosing condition [92]. The pooling of venous blood with increased venous wall pressure generates an exaggerated pro-oxidative environment. As a matter of fact, varicose veins are an example of chronic OS and iron deposition [90] which further enhance local inflammation and metalloprotease secretion with the consequent disintegration of the dermis [93,94]. VLUs are characterized by repeated cycles of healing and recurrence in 60% to 70% of patients [28,95]. The median recurrence time is 42 weeks with a recurrence incidence of 22% within three months after re-epithelialization, 39% at the sixth month, and 57% by the twelfth month [95]. The main risk factors for recurrence include a history of deep vein thrombosis, history of multiple previous ulcers, and longer duration of the previous ulcer [96]. It is accepted that further knowledge is required to fully identify and understand VLU recurrence drivers [88].

## 4. Common Pathologic Hallmarks in Chronic Wounds

Table 1 summarizes the major hallmarks of these three types of CW. Despite the differences in etiologies, pathophysiological constituents, and underlying mechanisms among the most common CWs, they share three major interacting and cooperative pathological hallmarks, which interfere with the healing process and may predispose patients to recurrence.

Chronic Inflammation: Inflammation is an essential, non-specific, innate immune response that, when prolonged in time and/or intensity, represents a major hallmark of non-healing wounds. This is the phase where CWs are frozen [101] and that may act as a proximal trigger for subsequent molecular derangements such as enhanced degradation of the GT matrix, impaired re-epithelialization, downregulation of growth factor activity, excessive ROS/OS cytotoxicity, and senescence in fibroblasts and other cells [102]. It must be emphasized, nevertheless, that the wound microenvironment inflammatory reaction is another finely regulated process in order to keep a homeostatic balance among critical interacting actors, such as innate immunity, pathogens/cell debris, release of damage- and pathogen-associated molecular patterns (DAMPs and PAMPs, respectively), and the amplitude of the host inflammatory response [103]. Critical cellular orchestrators of inflammation are cells of the innate immune system, such as neutrophils and macrophages. Neutrophil influx presides over the initial inflammatory response and they undergo apoptosis and are phagocytosed in the wound site in approximately 24 h, once the area is cleared from debris and pathogens. The diabetic ulcer environment predisposes to inflammation. Although the underlying mechanism remains elusive, diabetes stimulates neutrophil recruitment to the wound bed such as the excessive formation of neutrophil extracellular traps (NETs) by sustained activation of the NOD-like receptor protein 3 (NLRP3) inflammasome [104]. Converging evidence confirms that NETs are implicated in tissue damage and impaired wound healing [105,106]. Neutrophils isolated from the blood of DFU patients show an increased spontaneous NET formation which, in addition to high glucose levels, is driven by proinflammatory cytokines such as tumor necrosis factor alpha (TNF-α). To our view, a remarkable and recent finding is that elevated serum levels of NETs are a serological prognostic marker for the risk of DFU-related amputation [105]. Another pathogenically relevant event for wound chronicity is the failure of transition in macrophage polarization, which accounts for an exaggerated infiltration and protracted residence of proinflammatory M1 phenotype cells. This causes a surge in proinflammatory cytokines in the wound bed. Cyclooxygenase 2/prostaglandin and kallikrein-binding protein appear to drive M1 perpetuation through nuclear factor kappa B (NF-κB) signaling, at least in the DFU milieu. In the prevailing M1 macrophage context, the pool of growth factors involved in productive GT development (vascular endothelial growth factor (VEGF), fibroblast growth factor (FGF), insulin-like growth factor-I (IGF-I), transforming growth factor beta 1 (TGF-β1)) and secreted by the M2 subpopulation is significantly reduced. Concurrently, angiogenesis may be aborted or morphologically abnormal. Furthermore, related to subclass M1-mediated inflammation is the local exacerbation of matrix metalloproteinase (MMP) activity, promoting collagen degradation and hindering matrix reconstitution [98,107]. Since the failure of the transition to a prohealing M2 macrophage subclass impairs the normal healing process [108], restoration of macrophage polarization has turned into an attractive research target [109]. Exaggerated inflammation also characterizes VLUs and PI [110]. Biopsies of non-ulcerated skin from CVD patients (without active VLUs) show increased inflammatory infiltration, mostly by macrophages and T-lymphocytes, suggesting that perpetuated inflammation in CVD precedes the onset of the ulcer [111]. In correspondence with this finding, a macrophage subpopulation with an M1 proinflammatory phenotype characterized by TNF-α and hydroxyl radical production was described, which co-existed with resident fibroblasts having a p16INK4a-dependent senescence program. This was a seminal study toward the identification of the pathogenic role of a persistent M1 macrophage subpopulation as a healing deterrent factor in VLUs [112]. The subepidermal iron overload typical of VLUs has been pointed out as a perpetual signaler for monocytes–macrophages and the subsequent shift toward an M1 phenotype. Similar findings were obtained in relation to the level of proinflammatory cytokines in non-healing VLUs, where the cytokine levels progressively decreased as the ulcers healed [113]. Another illuminating study has recently demonstrated the pathogenic involvement of specific sequences of the human endogenous retrovirus genome, as a driver of inflammation and of a local immune pathological response in CVD and potentially in ulcer onset [114]. In general terms, CWs are highly susceptible to bacterial colonization and infection [115], thus leading to a sustained, although inefficient, battle between the innate immune activation and the biofilm, hitherto amplifying inflammation [116]. The wound microbiome and the formation of bacterial biofilms contribute to delayed wound healing by implementing a persistent inflammatory program [117,118].

Cytotoxicity by Oxidative Stress: It remains to be understood how the chronicity programs begins and progresses [119]. However, far from defining a hierarchal pathogenic cascade, it is acknowledged that, along with inflammation and the onset of societies of senescent cells, OS is a consubstantial hallmark for CWs [120,121]. Under physiological conditions, 90% of endogenous oxygen radicals are produced by mitochondrial oxidative phosphorylation and other electron transfer reactions. Partially reduced and highly reactive O_2_ metabolites, including superoxide anion (O_2_•−), hydrogen peroxide (H_2_O_2_), and hydroxyl radical (•OH), are originated within cells [122]. These oxygen-derived reactants are collectively identified as ROS, and all seem to play an essential role in cell metabolism, including their signal transduction ability throughout the healing stages. In the tissue repair context, intracellular ROS modulate the interaction between growth factors, cytokines, and chemokines and their respective receptors [123]. Thus, even in normal circumstances, inflammatory cell chemotaxis, recruitment, and cytokine secretion are influenced by ROS. Furthermore, free radicals and ROS participate in the first line of defense against wound-invading pathogens, extracellular matrix synthesis, and angiogenesis regulation [124,125]. In general terms, ROS have been evolutionarily integrated into multiple protection and metabolic/adaptative programs [126]. Relevant to wound biology, and specifically to diabetic ulcers, is the interaction of ROS with membrane fatty acids leading to an oxidative decomposition of the cell lipid radicals that, in turn, react with oxygen to generate highly cytotoxic lipid radicals. This process is known as lipid peroxidation [127]. In eukaryotic cells, a set of antioxidant molecules and a group of endogenous antioxidant specialized enzyme systems ensure a finely balanced environment for cell survival [126,128]. However, ROS homeostasis disruption by an imbalance between oxidant and antioxidant forces, favoring the oxidants, leads to so-called OS with the ensuing alteration in metabolic signaling, transcriptional processes, and massive oxidative damage to macromolecules [127]. In these circumstances, ROS levels may become lethal for wound cells, particularly fibroblasts and keratinocytes [129]. The mechanisms whereby ROS and OS increase within the CW environment are variegated, complex, and may arise from multiple intrinsic and extrinsic sources. However, the phase of inflammatory response to injury appears as the critical temporary event for ROS generation and spillover [117]. Low-level, chronic exposure to ROS accelerates telomeric shortening; in contrast, high levels of ROS induce telomere-independent premature senescence and apoptosis, which seem to rely on c-Jun N-terminal kinase (JNK) pathway activation. Accordingly, ROS increases fibroblast apoptosis, premature senescence, and poor growth response even in the absence of senescence markers, all included under the concept of “diseased fibroblasts” [130]. In a kind of positive feedback loop, these “diseased fibroblasts” entrenched in CW are a significant local source of ROS as compared with normal dermal fibroblasts [120]. Remarkably, this process of ROS-mediated exacerbation of telomeric dysfunction and cell senescence is intensely accelerated by systemic chronic inflammation and contrariwise inhibited by anti-inflammatory or antioxidant pharmacological interventions [131]. The experimental inhibition of some of these oxygen reactants in murine models of diabetic wounds were shown to restore fibroblast and keratinocyte physiology, resuming the normal healing trajectory [132]. Again, diabetes mellitus turns into an exemplary pathology in which ROS/OS play a proximal pathophysiological role in disease complication progression and irreversibility [133]. Under glucotoxic stress, upregulated biochemical pathways such as polyol, protein kinase C (PKC), and hexosamine drive in excessive generation of OS, along with inflammatory mediators which create a cytotoxic environment in GT cells [134]. An elegant study by Martins-Green’s group aimed to establish the hierarchal sequence of molecular damage driving the onset of wound chronicity. The study showed that proximal to high ROS spillover is the downregulation of nuclear respiratory factor 2 (Nrf2) as a critical transcription factor in redox homeostasis and mitochondrial preservation and biogenesis [135]. They also described a collapse of the phosphatidylinositol 3-kinase (PI3K)/AKT1 salvage pathway via the phosphatase and tensin homolog (PTEN) inhibitory effect, which represents a depletion of major cytoprotective mechanisms. Relevant to OS is the finding that proinflammatory lipids derived from arachidonic acid metabolism are increased with a concomitant reduction in pyruvate production, which may translate to reduced ATP production and wound arrest [119]. NADPH oxidase and xanthine oxidoreductase are both ROS-generating enzymes associated with respiratory burst and appear upregulated in professional phagocytic cells within the CW bed [136,137,138]. Inflammatory cytokines like TNF-α contribute to increased mitochondrial ROS generation and cell death. In turn, mitochondria-derived ROS elicit apoptotic death in a TNF-dependent manner by inducing a ROS-dependent activation of JNK. ROS are also known to activate Fas receptors and other death receptors [139]. In the line of inflammation and OS is the observation that primed neutrophil generation of NETs is a ROS-dependent process [140]. As mentioned for the pharmacological manipulation of the inflammatory reaction, the potential restoration of the redox balance within the wound microenvironment is a promising therapeutic avenue [141].

Cellular Senescence: Increasing evidence shows that cell senescence plays a significant role in the development and perpetuation of the chronicity phenotype of the different forms of CW. It has been reported that 15% of senescent cells is the limiting threshold from which wounds adopt a refractory-to-healing phenotype. Furthermore, the entrenchment of a senescent cell population in CWs is a major and persistent factor for an inflammatory environment, with the release of proinflammatory cytokines and inhibition of proliferation of GT-building cells [142]. Consequently, preventing cell senescence or removing senescent cells is projected as a new therapeutic strategy for problem wounds [143]. Senescence is a cellular program that induces proliferative arrest accompanied by morphological modifications, metabolic reprogramming, implementation of a complex proinflammatory spreadable secretome, increased autophagy, apoptosis resistance, and epigenetic re-editing [144]. In general, this is a cellular response to multiple types of stress, such as DNA damage, telomere erosion, oncogene activation, oxidative damage, protein misfolding, and exposure to extracellular signals, which may occur at any point in the cell’s life cycle [143]. Relevant for the wound microenvironment, OS appears as a proximal trigger of cellular senescence, with ROS acting as signaling molecules for the so-called stress-induced premature senescence [6]. Senescent fibroblast populations have been identified and characterized in biopsies of human CWs, consistently associated with impaired healing [6,145]. At the molecular level, p53 and p16INK4a/Rb are two senescence-regulatory hubs leading to proliferative arrest. Moreover, given the elevated and multilayered expression of p21Waf1/Cip1, it is considered a far more specific senescence marker of wound arrest [6]. These fibroblast populations are endowed with a cytotoxic senescence-associated secretory phenotype (SASP) that spreads senescence messages, along with matrix-degrading proteases and other inflammatory reactants. This evidence has been experimentally validated since increased skin senescent cells and SASP markers in young mice are associated with delayed wound healing [146]. It is likely that diabetic lower limb wounds are the most representative model of the link between wound chronicity and societies of senescent fibroblasts [44]. Telomeric shortening and dysfunction have been described in diabetes via proinflammatory cytokine production and ROS generation, which, in turn, creates a vicious circle between mitochondrial damage/dysfunction and ROS production [147]. In this scenario, high glucose levels seem to act as a primary damaging factor for mitochondrial populations, leading to reduced biogenesis, abnormal fission, organelle fragmentation, and malfunctioning with massive effusion of ROS [148]. As a corollary, diabetic patients with ulcers present lower telomerase activity in their leukocytes as compared to patients without ulcers, irrespective of the duration of diabetes, which may indicate a systemic repercussion of the prosenescence secretome originated in the wound [149]. In line with this, interesting experiments have shown that skin senescent cells can spread the senescence message to distant organs, thereby accelerating systemic aging processes and deteriorating physical and cognitive functions [150] as is described for diabetes [151]. Senescent fibroblasts are critical pathophysiological actors for pressure [152] and venous ulcers [153]. In these wounds, fibroblasts become prematurely senescent, secrete prosenescence messages, and, like diabetic fibroblasts, conserve a refractory-to-proliferation memory when cultured, which seems to be due to a failure in DNA replication [154]. Microarrays and functional analysis of pair-wise comparisons of telomere dynamics between normal fibroblasts and chronic-wound-derived fibroblasts (CWFs) indicate that CWFs have a decreased ability to withstand OS, which may be a driving factor toward cells’ premature senescence. Ultimately, the study concluded that, although the persistence of non-healing wounds is in part due to prolonged chronic inflammation and bacterial infection, premature fibroblast senescence is implicated in disease chronicity [155]. Another recent study has confirmed that cellular senescence and downregulation of the immune response were represented by the common shared dysregulated sets of genes in representative murine models of aged, diabetic, stagnant healing as compared with normal young mice [156].

Conclusively, the wound is a vulnerable substrate in which converging factors like inflammation, OS, and cellular senescence act as leading primers toward a chronicity phenotype (Figure 4). These three pathogenic pillars progress together and trigger and amplify one to another, ultimately deterring the repair process. To our view, one of the most meaningful aspects of this pathological landscape is the downfall of the agonistic axis among the growth factors and their receptors [157], which may ultimately delay matrix protein synthesis and secretion, impair angiogenesis, reduce fibroblasts’ tolerance to stress infection and survival, and perpetuate cellular senescence [38].

## 5. Chronic Wound Cells May Harbor a Chronicity–Recurrence Epigenetic Memory Code

Years ago, the hypothesis on the existence of an “open portal” memory as a theoretical explanation for the high rates of chronicity and recurrence of chronic wounds was established. It was accordingly assumed that this memory underlies a *de novo* edited epigenetic code that imposes abnormal pathologic behaviors of dermal fibroblasts and keratinocytes as memory cells. We conceive that these endogenous healing-deterrent factors are represented by soluble circulating signalers, such as by in-scar anchored societies of dysfunctional fibroblasts that, through their secretome, create and perpetuate a paracrine cytotoxic environment, which in cooperation with external predisposing factors leads to lesion stagnancy and recurrence [30]. The recent advent of novel technologies for single-cell genetic and epigenetic dissection allowed for charting gene clusters and epigenetic signatures involved in normal and pathologic tissue healing, including wound chronicity [98,158]. The list of epigenomic alterations and their significance in wound pathology is continuously escalating, providing a foundation to our hypothesis [159]. Metabolic stress and hypoxia exposure prime genes to have memory. The most illustrative example is that of hyperglycemia leading to various persistent epigenetic changes, including DNA methylation, histone methylation, and histone acetylation, provoking sustained activation of proinflammatory pathways as well as oxidative stress [160]. This epigenetic-based cellular memory through the editing of an abnormal signature ensures the onset and perpetuation of the chronic wound phenotype [161,162]. Hyperglycemia may increase DNA mutations, DNA breaks, genomic instability, and particularly epigenetic dysregulation. Accordingly, hyperglycemia and a constellation of downstream factors transform the cellular native epigenetic architecture, rendering a de novo code that largely alters cellular physiology [163]. Impaired diabetic healing is largely driven by an epigenetic program that pushes the cells along a slippery slope of proliferative arrest, precocious senescence, persistent inflammation, and OS, ultimately dismantling the intrinsic drivers of a physiologic repair process (Figure 4). This glucotoxicity-derived program establishes the foundations of the metabolic memory and safeguards its persistence, progressivity, and transfer through generations of cells. This explains why prior hyperglycemic stress is not “forgotten” and why successive generations of descendant fibroblasts collected from the wound bed exhibit a persistent abnormal response under ideal culture conditions [50,160,164]. The pioneering findings by Assam El-Osta inaugurated a new era by showing that exposure to simulated hyperglycemia was sufficient to reprogram cells’ native epigenetic landscape and that these modifications largely persisted even beyond the normalization of glucose levels. This hyperglycemia-derived effect was associated with changes in the methylation pattern in the promoter region of genes controlling inflammation [165] and OS [166]. The transcendental essence of El-Osta’s findings is that, for the first time, the controversial concept of metabolic memory received a solid mechanistic foundation at a molecular level: short-term exposure of cells to high glucose stress is sufficient to orchestrate a long-lasting inflammatory program based on an epigenetic blueprint. Subsequent studies have further validated and broadened the concept that, behind the torpid healing phenotype in diabetes, there is an abnormal cellular memory that is transmitted through generations of wound cells that may linger within the residual scar and that, in the end, may predispose to recurrence [30,167,168]. There are multiple descriptive examples and models which show how diabetic metabolic memory disrupts wound cell physiology and primes the wound toward the onset of a chronicity phenotype through the interplay among local inflammation, oxidative stress, and cell senescence [23,169] (Figure 5). Furthermore, this abnormal epigenetic landscape seems to impair dermal cell and keratinocyte topographic memory and positional identity [162,170].

### 5.1. In Vivo Transmissibility from Humans to Rats of Diabetic Wound Markers

GT is a short-life, transient, contractile organ, evolutionarily responsible for filling a gap or being a tissue welder, characterized by the presence of proliferative, contractile, and secreting fibroblasts/myofibroblasts, active angiogenesis, and inflammatory cell infiltration within its extracellular matrix [2]. We have recently described a set of angiogenesis pathologic traits in diabetics’ GT biopsies. Accordingly, from the early stages of organization, GT emerging arterioles exhibited a remarkable wall thickening, endothelial inflammation, and a luminal obliteration that frankly characterizes and evokes diabetic microangiopathy [25], a process that is not ordinarily implemented in a period of days [171]. We therefore hypothesized that angiogenesis-committed cells and other endothelial progenitor cells were “born” with an imprinted pathologic angiogenesis program attributable to diabetes. Thus, the next experiment of our group leveraged the ancient legacy of Francis Peyton Rous so that cell-free filtrates (CFFs) were elaborated using diabetic-subject-derived samples of GT and limb arteries and nerves collected during amputation surgery. Our premise was based on the fact that the CFF could act as a vector to transfer soluble signalers which may impose the diabetic donor’s histological hallmarks on otherwise healthy rats. The material was injected into full-thickness, controlled wounds once a day for seven days. Primarily, the diabetic homogenate was proven to arrest the healing process by reducing GT growth and contraction rate. GT histopathological analysis revealed the presence of microangiopathy and neuropathy with acute onset as diabetes-typical hallmarks and that largely mirrored the donors’ tissue histopathology (Figure 6) [52]. These data suggested that diabetes “damage memory” in the human donor tissues can be transmitted to other mammal species and impose the “typical damage phenotype”. This study also showed that these diabetic damage drivers are present both within the ulcer and in distant, internal tissues [52]. None of these findings was identified in the corresponding controls, including a group of rats exposed to large amounts of AGEs (683.8 ng/mg of protein).

Fibroblasts are cells with a large reserve of plasticity and epigenetic reprogramming before external clues, which may alter their biological behavior and, ultimately, wound-healing fate. Most importantly, fibroblasts are endowed with the ability to retain a memory from their positional location, mechanical, chemical, and inflammatory signals [172,173]. Classic findings from more than 20 years ago, as well as other more recent ones, document that fibroblasts collected from diabetic and non-diabetic CWs exhibit and transmit in culture phenotypic traits as if they remain within the donor’s ulcer. Thus, primary cultured fibroblasts and descendent cells are characterized by reduced proliferative life span, senescence or increased susceptibility to senescence, reduced ability to withstand OS, inability to correctly express a stromal address code, as well as morphological alterations when compared to normal dermal fibroblasts [155,174,175,176]. Interestingly, the conditioned medium from these “diseased cells” is able to transmit and impose the above-mentioned abnormal traits to normal cultured fibroblasts [177]. All these studies suggest the existence of a fibroblast’s pathologic memory or, at least, that a senescence-like memory is retained and transmitted in vitro as a part of a putative CW epigenetic signature [178]. How this fibroblast memory is steadily maintained once the cells are extracted from the donor and how this memory articulates with wound recurrence are subjects of intense research. Recent data confirm the protagonist role of epigenetic drivers in fibroblast plasticity, reprogramming, and wound fate [169,172,173].

### 5.2. Epigenetic Drivers in Wound Chronicity

DNA methylation and non-coding RNAs (ncRNAs) represented by microRNA (miRNA), long non-coding RNA (lncRNA), and circular RNA (circRNA) are the main epigenetic mechanisms involved in normal and pathologic wound healing described so far [98] (Table 1). DNA methylation appears to be the most abundant epigenetic modification and is involved in the regulation of physiological processes like inflammation, proliferation, migration of keratinocytes, angiogenesis, and MMP secretion [179]. Aberrant DNA methylation is a trait associated with wound chronicity which has been identified in diabetic and non-diabetic ulcers. An unbiased whole-genome methylome of CW edges determined that, at CW margins, DNA hypermethylation predominated versus DNA hypomethylation which inhibited epithelial–mesenchymal transition and was proven to impair re-epithelialization [159]. Fibroblasts from non-healing DFUs show lower global DNA methylation, while functional annotation identified enrichment of genes associated with angiogenesis and ECM assembly [180]. Direct evidence of AGEs impacting DNA methylation in diabetes was described. AGE-BSA contributed to upregulating MMP-9 transcription via ten-eleven translocation 2, a well-known DNA demethylation protein that activates MMP-9 promoter demethylation in diabetic skin tissues and primary keratinocyte cultures [181]. The Wilms tumor 1-associated protein (WTAP)-DNA methyltransferase 1 (DNMT1) axis acts as a pivotal regulator of diabetic foot ulcer healing. WTAP is upregulated under high-glucose conditions, leading to an excessive expression of DNMT1, ultimately impairing angiogenesis, reducing cell viability, and delaying wound contraction. Conversely, WTAP knockdown in a diabetic mouse model restores endothelial function and significantly improves wound healing [158]. Histone modifications have also been observed in diabetic wounds, so that DFUs exhibit higher levels of HDAC2 expression in their endothelial progenitor cells. Inhibitors of this HDAC enhance wound healing by encouraging endothelial cell proliferation and tube formation [182]. In addition to DNA methylation and histone modifications, other epigenetic drivers like non-coding RNA have been described as major actors in CW memory [183]. MicroRNAs (miRNAs) are short, highly conserved, non-coding RNA molecules (~22 bp) that regulate gene expression at a post-transcriptional level [184,185]. MiRNAs are relevant players in wound-healing physiology by regulating inflammatory signaling pathways and modulating the function of immune cells. Mounting evidence documents an existing miRNA deregulation in chronic wounds, which contributes to their chronicity by perpetuating inflammation and limiting fibroblast and keratinocyte migration. Accordingly, intense contemporary research focuses on the potential of microRNAs as diagnostic tools for chronic wound clinical management [186]. Some specific miRNAs that are reliable plasmatic candidates as predictors of ulcer inflammation, size, chronicity evolution, and even patients’ psychological distress include miR-191, miR-200b, miR-21-5p, miR-155-5p, miR-146a-5p, and miR-221-3p [187]. Although the studies addressing the pathogenic participation of miRNAs in VLUs and PI are limited, it is recognized that they impact CW biology by limiting cell migration, angiogenesis, extracellular matrix synthesis, wound contraction, remodeling, and by conversely enhancing the inflammatory responses and oxidative stress [68,100]. Opposing scenario examples exist about miRNA whose upregulation (miR-24) is negatively correlated with amputation rate in DFUs and positively correlated with healing response. Low plasma levels of miR-24 in diabetic subjects was closely related to the occurrence, development, and prognosis of DFUs and osteomyelitis, being considered as a potential biomarker for the prediction of DFUs [188].

While a small portion of the mammalian genome is transcribed into protein-coding mRNAs, the majority of the genome is transcribed into lncRNAs [189] which are RNAs longer than 200 nucleotides not translated into proteins [190], with an important role in gene expression including the wound-healing process [191]. Globally speaking, lncRNAs are involved in controlling wound inflammation, macrophage polarization, lymphatic angiogenesis, and keratinocyte migration and differentiation. For the latter, studies converge to indicate that lncRNAs regulate keratinocyte progenitor proliferation and differentiation [192]. RNA-sequencing studies have identified a multitude of differentially expressed lncRNAs in diabetic wounds compared with non-diabetic wounds. Most importantly, some lncRNAs have demonstrated a predictive or prognostic value within the DFU realm. An example is upregulated lncRNA-H19, which is detected in diabetic blood samples while its circulating levels increased in parallel to wound inflammation [193]. Similarly, lncRNA-GAS5 levels are higher in diabetic wound skin samples than in non-diabetic counterparts. Upregulation of GAS5 in macrophages is associated with diabetic ulcers’ chronic inflammation and prolonged subclass M1 macrophage polarization [194]. Lnc-RNA MRAK052872 (named lnc-URIDS) is dramatically increased not only in the diabetic skin but also in the serum of type 2 diabetic patients with DFUs, reducing granulation tissue formation and consolidation [195]. Contrariwise, cutaneous PDGFRα+ fibroblast-expressing lncRNA-H19 (lncH19) accelerates the wound-healing process via promoting dermal fibroblast proliferation and macrophage infiltration [196]. Preliminary data indicated that lncRNA metastasis-associated lung adenocarcinoma transcript 1 (MALAT1) silencing contributed to reducing proinflammatory cytokines TNF-α and IL-6, while the anti-inflammatory cytokine IL-10 was increased, suggesting that MALAT is associated with the regulation of inflammation [197]. In a similar manner, the so-called “wound and keratinocyte migration-associated long non-coding RNA 2” (WAKMAR2) is also involved in the regulation of inflammatory chemokines produced by keratinocytes [198]. It is likely that one of the most sound findings of lncRNA in wound physiology is from the study by Li and co-workers, who showed through a variety of molecular analyses that lncRNA SNHG26 promoted the transition from inflammatory to proliferative phases of keratinocyte progenitor cells. Mechanistically, SNHG26 interacted with ILF2, shifting this transcription factor from inflammatory genomic loci to the LAMB3 locus, thereby facilitating the switch from inflammation to re-epithelialization [199].

CircRNAs are important regulators in the cellular life cycle. They are a class of non-coding RNAs formed by covalent cyclization and are abundantly expressed in eukaryotic organisms and have high stability [200]. It has been recently shown that circRNAs encode proteins and may be involved in the regulation of gene transcription. Increasing evidence confirms the biological significance of circRNA in wound-healing biology, including tissue remodeling, fibroblast cycle progression and migration, senescence, angiogenesis, apoptosis, vascular protection, and cytoprotection [201]. CircRNAs are detected within the wound tissue and blood samples of DFU patients and their expression levels make them potential biomarkers of DFU onset, severity, and evolution. Illustratively, the expression of circ_0000907 and circ_0057362 is negatively correlated with the ankle–brachial index (ABI) and percutaneous partial pressure of oxygen (TcPO2). Furthermore, the serum levels of these circRNAs are positively correlated with the severity of DFUs, which may offer diagnostic/prognostic support to the patients’ classification and staging through the Wagner scale [202].

Although research on circRNAs and other forms of non-coding RNA is still in its infancy and its clinical validation may require time, funds, and technical accuracy, these non-coding RNAs may be powerful and reliable biomarker tools in predicting the course and outcomes of the healing process. At the same time, the existence and surreptitious involvement of an abnormal epigenetic landscape in the onset and persistence of a chronicity program contributes to validating our notion of “open portal memory”.

## 6. Concluding Remarks

This review has intended to discuss a major critical aspect of the intricate, dynamic, ordered, and overlapping process of wound healing: the phenomenon known as wound chronicity. The event is characterized by a sort of wound paralysis since adjacent or recruited cells in the injured site become insensitive to alarm signals and/or irresponsive to chemotaxis and proliferative cues. Thus, the terrestrial amniotes’ natural “response to injury” characterized by a fibroplasia induration partitioned by new growing vessels is torpid, heterogeneous, or merely arrested. After years of intense basic and translational research and huge financial investment, we are still far from understanding the mysteries behind why wounds fail to heal. A challenge beyond the elucidation of the endogenous triggers of wound chronicity is the need for charting the chronopathogram in which a hierarchal order of the pathologic events ruling chronification may be established. The episodes of wound recurrence and metastasis in the contralateral leg (in diabetic subjects) are a serious clinical and academic challenge. As a matter of fact, the number of cases of recurrence are frankly alarming. A pending task is the identification of endogenous recurrence predisposing factors and predictive biomarkers that could help to anticipate if the epithelial cover or the neodermal scar are at risk. We still have much to learn about the innate tools used by our body to repair injured tissues. The fact that, irrespective of differences in their individual pathophysiological cascade, DFUs, PI, and VLUs share major distal damage operators such as protracted inflammation, OS, and cell senescence has opened novel research avenues that may reveal future therapeutic opportunities. Current intense research lines are centered on: restoring macrophage subpopulation shifting, attenuating neutrophils’ excessive NETs, repositioning a physiological REDOX balance, identification of senolytic activators, preventing or reducing mitochondrial dysfunction, and understanding and manipulating wound cells’ metabolic programs. We advocate the notion that inflammation, OS, and the prevalence of senescent or diseased fibroblast populations are the distal consequences of an imprinted epigenetic memory that is not exclusive to diabetic wounds and that predisposes to chronicity in other types of ulcers. Two simple observations are: (1) PI and VLU cultured fibroblasts exhibit a short replicative life span, becoming irreversibly senescent and refractory to growth factor stimuli and, aligned with this, (2) the chronicity phenotype is restored over and over again despite the attempts to make the wound milieu “acute” with repeated debridements. We also hold the hypothesis that the “entrenchment” of fibroblast and/or keratinocyte societies within this type of open portal legacy predisposes to recurrence. In reciprocity, OS is a critical contributing force to sculpt and spread this kind of open portal epigenetic message. The commanding role of epigenetics in wound healing is not restricted to the problem of deficitary healing. Mounting recent evidence indicates that numerous epigenetic modifications such as DNA methylation, histone modification, microRNAs, and long non-coding RNAs play a definitive role in keloid recurrence as a prototype of exuberant healing. Manipulating this pathologic epigenetic code is a novel promising therapeutic window toward esthetic healing and keloid recurrence prevention. Accordingly, the possibilities offered by wisely intervening in epigenetic memory are limitless for CW pathology as well as for other diabetic complications. The list of validated epigenetic modifiers is progressively increasing. Although we envision a brilliant horizon for epigenetic medicine success, it still requires knowledge and being sharp, selective, and precise in playing the keys of the cells’ memory code.

## Figures and Tables

**Figure 1 ijms-26-08745-f001:**
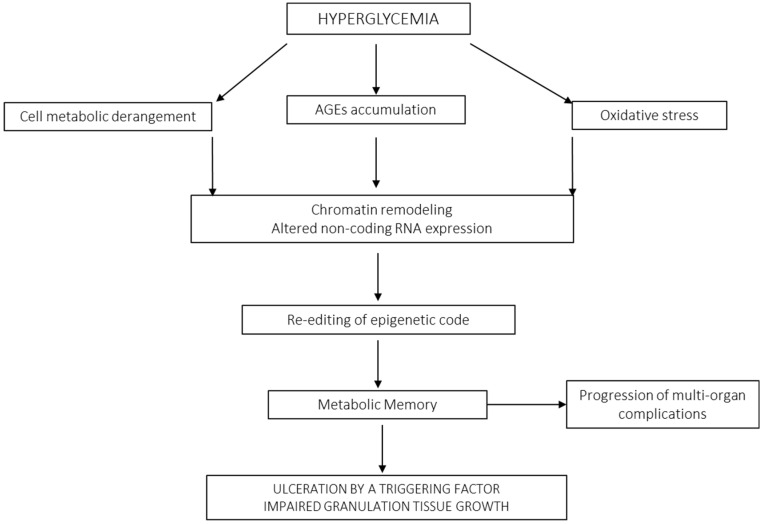
From “trivial” glucose levels to metabolic memory. High glucose levels disrupt cellular physiology and introduce abnormal pathways, leading to metabolic and transcriptional changes. AGEs accumulate in tissues and cells, being highly proinflammatory and cytotoxic. Hyperglycemia is a basic precondition for the generation of free radicals, OS, and depletion of antioxidant reserves. Hyperglycemia is associated with DNA damage, inducing nuclear and mitochondrial DNA mutations. Furthermore, glycated radicals stick onto histones, altering DNA transcription and editing a *de novo* epigenetic code which is the foundation of metabolic memory and therefore of all diabetic complications, including cutaneous and mucosa healing deficits. DNA is known to be a target of OS free radicals which also preconditions the establishment of the cellular senescence phenotype such as that seen in “diseased fibroblasts”. The molecular mechanistic bases of the perpetuation and transmission of the epigenetic code and therefore of the metabolic-memory-derived cellular behavior have not been elucidated in diabetes pathology. Inflammatory reactants and OS appear to contribute to the perpetuation of this epigenetic code.

**Figure 2 ijms-26-08745-f002:**
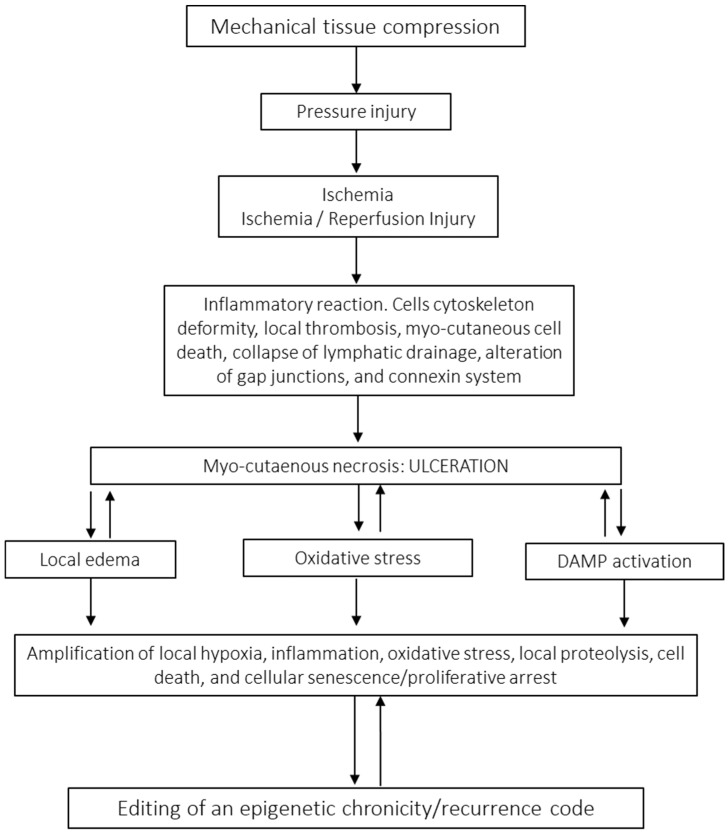
Putative pathophysiology of pressure injury. PI as a result of mechanical forces exceeding a tissue tolerance threshold which leads to ischemia and cellular hypoxia. In this process, myo-cutaneous necrosis is associated with the intrinsic metabolic demands of each tissue layer where the first inflammatory response is triggered. In addition to thrombosis and ischemic cell death, pressure destroys the system of cellular interconnection and the cytoskeleton. Under these circumstances the ulcer is definitively established, having a highly proteolytic biochemical milieu, which is further aggravated by the failure of lymphatic circulation which amplifies local edema and inflammation. Edema intensifies local hypoxia which in turn magnifies the pool of free radicals and the consequent OS. The exposure of intracellular self-antigens by the rupture of the gap junctions amplifies inflammation. This highly cytotoxic and free-radical-rich milieu may be a major driver for the onset of diseased fibroblasts in which proliferative arrest is part of the senescent program. We believe that this environment is permissive for the editing of an abnormal epigenetic code whose signature imposes proliferative arrest.

**Figure 3 ijms-26-08745-f003:**
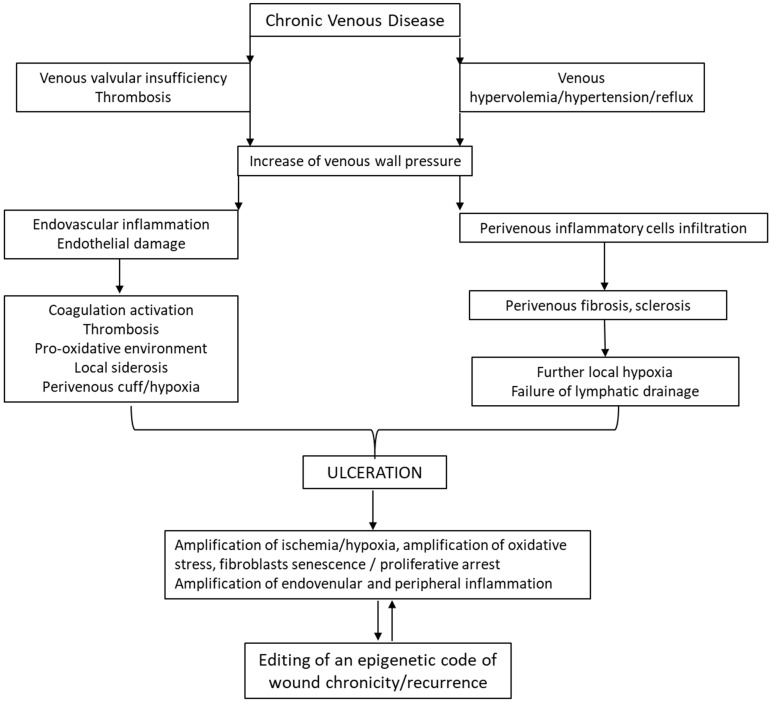
Venous ulcer pathophysiology. A venous ulcer is largely anticipated by a CVD in which inflammation and OS cytotoxic damage appear before skin necrosis and ulceration. Thus, this is a representative pathology of the pathogenic role of chronic inflammation and a pro-oxidant milieu in wound chronicity. Venous hypervolemia, valvular-failure-associated reflux, and the consequent intraluminal high pressure all converge to damage the endothelial layer and the perivenous space. These episodes are associated with further endo- and perivascular inflammation, thrombosis, and oxidative damage—ultimately leading to skin necrosis and ulceration. The mixture of arterial and venous blood is also implicated in this damage cascade in which hypoxia, OS, proteolysis, and inflammation are intensified. This is a cytotoxic environment for the local fibroblasts which become prematurely senescent and arrested. The in vitro behavior and other biomarkers in these venous-ulcer-derived fibroblasts suggest the existence of a driving epigenetic code of wound paralysis.

**Figure 4 ijms-26-08745-f004:**
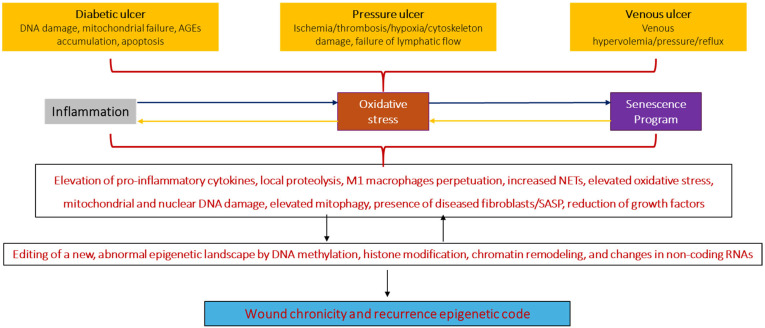
Common molecular drivers of chronic ulcers. Inflammation, OS, and the presence of senescent societies of fibroblasts are common features shared by diabetic, pressure, and venous ulcers; which at the same time are hallmarks of chronicity, irrespective of the differences existing in the etiopathogenic cascade. In this scenario the wounds become refractory to healing as the microenvironment is characterized by an imbalance between anabolic and catabolic activity, in favor of the latter. The GT-producing cells are embedded in a highly cytotoxic and inflammatory environment, enriched in ROS, which harms cells’ DNA, reducing the proliferative response and the mitochondrial respiratory activity with limited ATP production. There is a highly proteolytic activity which impairs matrix protein build-up and reduces the availability of growth factors and their receptors. The deficit of growth factors deters cell migration, proliferation, and protein synthesis and is a preconditioning factor for cellular senescence. We have accordingly hypothesized that this is a permissive environment for the re-editing of the epigenetic code through signalers like ROS, which may ultimately dictate the senescent and the arrest phenotype observed in cultured keratinocytes and fibroblasts derived from CWs.

**Figure 5 ijms-26-08745-f005:**
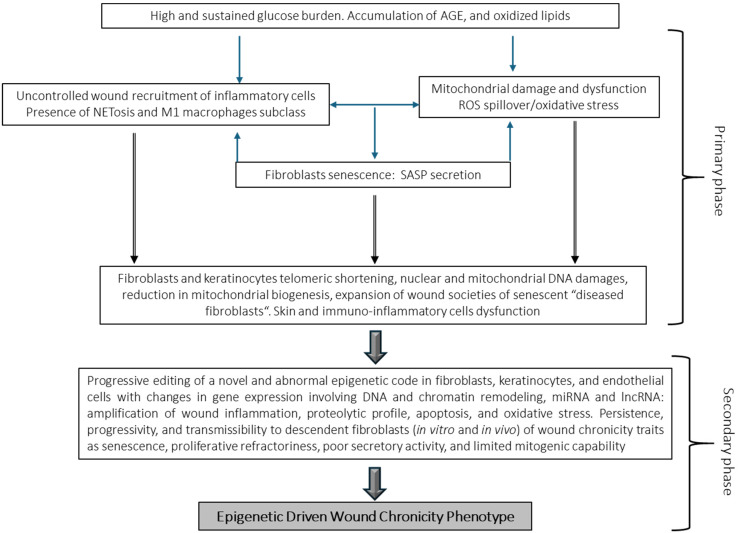
Interplay among inflammation, oxidative stress, and cellular senescence through epigenetic drivers in diabetes. The diabetic wound has been a model to illustrate how an epigenetic abnormal signature sculpted under the cytotoxic influence of hyperglycemia, AGE, and free radicals persistently activates genes leading to inflammation, oxidative stress, and cellular senescence. At the same time these three pathogenic actors, hallmarks of wound chronicity, create an interperpetuating crosstalk in what we hypothesize as the primary phase. Under this scenario the diabetic wound is densely infiltrated by polymorphonuclear cells with high NETosis rates or by subclass M1 macrophages with a proinflammatory activity. The failure of transition to subclass M2 macrophages is invoked as a major cause of wound chronicity. High glucose, AGE, and other cytotoxic products introduce nuclear and mitochondrial DNA damage which synergize to generate free radical spillover, further hampering mitochondrial ATP generation. The combination of these factors contributes to establishing and disseminating foci of senescent fibroblasts which are refractory to proliferation, generate more free radicals and inflammatory mediators, are somewhat hypoxic, and are prone to spread senescence messages to neighboring cells via extracellular vesicles. Under these circumstances a large population of irreversibly “diseased fibroblasts” is established with the orchestration of senescence and apoptosis programs that impair keratinocyte migration and polarity, thereby limiting not only granulation tissue growth but re-epithelialization. In a more advanced stage or secondary phase, the hallmark is the editing of an epigenetic code by different wound cells characterized by DNA and histone methylation, demethylation, histone acetylation, and transcriptional alterations with large lncRNA and miRNA which are the foundations of metabolic memory. The first distinct hallmark is that metabolic memory promotes persistence, progression, and transmission of harmful effects, including inflammatory changes, premature cell senescence, and apoptosis. Consequently, these epigenetic modifications contribute to the stable adverse effects of early hyperglycemic conditions on cellular functions even after subsequent normoglycemia. This situation is further complicated if the wound is ischemic since fibroblasts in hypoxic environments are also capable of generating an epigenetic metabolic memory.

**Figure 6 ijms-26-08745-f006:**
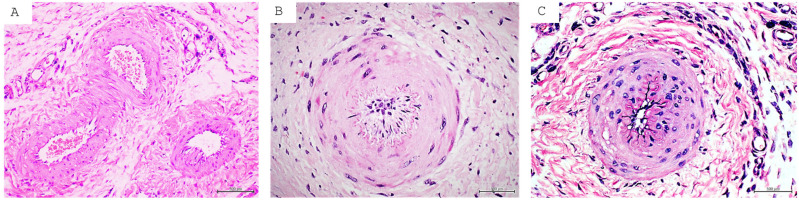
Diabetic microangiopathy reproduction in non-diabetic rats. (**A**) Image representative of arterioles within the GT of rats injected with healthy human donor normal GT homogenate. Note the normal morphology and the luminal aspect of three arterioles. (**B**) Histological image of arteriolar wall thickening from very early stages of the angiogenic response. A loose fibrotic matrix infiltrated with inflammatory cells and fibroblast-like cells is noticed. This image is representative of the microangiopathy process recreated in healthy rats whose wounds were infiltrated with a homogenate elaborated with GT of the ischemic ulcer of a diabetic subject. The fusion of thickened media and intima layers with luminal narrowing is noticeable. (**C**) Representative image of arteriolar wall thickening through media and intima layer engorgement and fusion with luminal narrowing in the GT of the ischemic diabetic ulcer used to prepare the homogenates. For all three panels: hematoxylin/eosin (H/E), magnification ×20, scale bar represents 500 µm.

**Table 1 ijms-26-08745-t001:** Most common human chronic wounds and their major characteristics.

Wound	Clinical Hallmarks	Recurrence Rates	Relevant Epigenetic Driver
DFU	Consequent to long-term hyperglycemia and cytotoxic by-products like AGEs and ROS. Ulcer onset by external factors like trauma, funguses, etc. Major cause of amputation and disability. Two main clinical classifications: ischemic and neuropathic. GT histopathology with large differences between the two clinical forms. The former with poor and abnormal angiogenesis, the latter with extensive nerve fiber edema and axonal degenerative changes.	About 40% of patients will experience recurrence within the first year of re-epithelialization. The wound is preferentially considered in “remission” instead of healed [31].	Exhibits the broadest and deepest characterization of epigenetic abnormal codes, introduced by hyperglycemia, ROS, and AGEs. This epigenetic code is the foundation of metabolic memory which sustains the perpetuation of complications, including the poor healing response in skin and mucosa. The epigenetic anomalies supporting inflammation, senescence, and oxidative stress have been identified. They include changes in DNA methylation pattern, histone modifications, and non-coding RNA [97,98].
PI	Largely prevailing wound in both elder and non-elder subjects mostly with spinal cord injury. Although with endogenous and external predisposing factors, the triggering one is prolonged pressure, friction, or shearing force over a bone prominence. It is a cause of multimorbidity and mortality in elder populations. At the histopathological level, the main pressure targets are large dermal vessels, leading to muscle necrosis and a cascade of damage concluding in full-thickness skin necrosis and adjacent vessel thrombosis.	The incidence of recurrence may span 5.4–73.6%. The most commonly reported site of recurrence is the ischium [74].	Epigenetic studies on PI are scarce. To our understanding a major and recent contribution is the identification of genes related to recurrence. Upregulated activity in genes involved in metabolic pathways (ENOSF1) and pathways in biological senescence (TMEM158) and downregulated activity in the interleukin 17 signaling pathway are directly involved in antimicrobial protection in vivo. Conversely, persons without recurrent PIs have upregulated activity in genes related to bioprocesses such as regulators of vasodilation (TAC3) and downregulated activity in biological pathways related to addiction (FOS, JUN, FOSB) [99].
VLU	Largely prevailing chronic wound consequent to long-term chronic venous disease. It imposes a poor quality of life. VLUs exhibit a complex and intricate pathophysiological mechanism, involving hemodynamic, cellular, and molecular alterations of macro- and microcirculation, which ultimately drives skin necrosis by venous ischemia, hypertension, and hypoxia. Histological examination reveals dilated veins, hemosiderin and fibrin deposition, sclerosis hemorrhage, and vascular fibrin cuffs. Acute phase lipodermatosclerosis shows lymphocytic and inflammatory cell infiltrate.	VLUs are characterized by repeated cycles of healing and recurrence in 60% to 70% of patients [28,94]. The median recurrence time is 42 weeks with a recurrence incidence of 22% within three months after re-epithelialization, 39% at the sixth month, and 57% by the twelfth month [95].	Numerous upregulated miRNAs have been shown to participate in prolonging VLU inflammation and limiting angiogenesis. Concurrently, the involvement of upregulated miRNA inhibited keratinocyte proliferation, prolonged inflammation, and impaired the healing response [100].

DFU: diabetic foot ulcer; PI: pressure injury; VLU: venous leg ulcer.

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
