# Peer review of "Major Common Hallmarks and Potential Epigenetic Drivers of Wound Chronicity and Recurrence: Hypothesis and Reflections"

_ijms, 2025, doi:10.3390/ijms26178745_

Round 1

Reviewer 1 Report

Comments and Suggestions for Authors

The manuscript “Major Common Hallmarks and Potential Epigenetic Drivers of Wound Chronicity and Recurrence: Hypothesis and Reflections” presents a comprehensive review of chronic wound pathophysiology, with emphasis on diabetic foot ulcers (DFU), pressure injuries (PI), and venous leg ulcers (VLU). It identifies chronic inflammation, oxidative stress, and cellular senescence as common hallmarks and proposes that an epigenetic memory in fibroblasts and keratinocytes may drive both chronicity and recurrence. The article synthesizes current literature, highlights gaps in understanding, and provides a conceptual framework for future research on epigenetic modulation in wound therapy.

  1. Consider adding a brief graphical abstract or schematic summarizing the interplay between inflammation, oxidative stress, cellular senescence, and epigenetic memory.
  2. Some sections are long and dense (e.g., the description of pathophysiological mechanisms in Sections 3 and 4). Adding subheadings or short summary boxes would improve readability.
  3. Ensure consistent use of abbreviations: for example, OS (oxidative stress) and CW (chronic wound) should be defined once and used consistently.
  4. Consider adding a summary table comparing DFU, PI, and VLU in terms of: Hallmarks, Recurrence rates and Epigenetic signatures.
  5. Minor grammatical edits for conciseness are recommended. Examples:

“…represents a challenging investigational and social problem…” → “…represents a clinical and research challenge…”

“…they may appear commonly granulated but not re-epithelialized…” → “…they often present with granulation tissue but lack re-epithelialization…”

Author Response

Reviewer 1.

Havana, August 16, 2025

Dear Sir/Mme.

Once again, I would like to express you my gratitude for the excellent critical reviewing of our manuscript and the corresponding proposals, which we have respected and followed one by one.  I feel that though your constructive criticism, the manuscript will be improved.  The changes could be observed through the side-comments introduced all across the text of the manuscript. 

Thank you very much.

Jorge Berlanga-Acosta. Senior Author. 

Please, find below the changes introduced as per your recommendations.  

  1. Consider adding a brief graphical abstract or schematic summarizing the interplay between inflammation, oxidative stress, cellular senescence, and epigenetic memory. This is undertaken. Represented in figure 6.
  2. Some sections are long and dense (e.g., the description of pathophysiological mechanisms in Sections 3 and 4). Adding subheadings or short summary boxes would improve readability.  Few sub-headings were introduced and the information that could be provided through summary boxes is summarized in the table proposed as to compare each type of ulcer. The length of the manuscript has been expanded.
  3. Ensure consistent use of abbreviations: for example, OS (oxidative stress) and CW (chronic wound) should be defined once and used consistently. Revised.
  4. Consider adding a summary table comparing DFU, PI, and VLU in terms of: Hallmarks, Recurrence rates and Epigenetic signatures. This is undertaken. Represented in table 1.

Minor grammatical edits for conciseness are recommended. Examples:

“…represents a challenging investigational and social problem…” → “…represents a clinical and research challenge…” – Accomplished

“…they may appear commonly granulated but not re-epithelialized…” → “…they often present with granulation tissue but lack re-epithelialization…” - Accomplished

Reviewer 2 Report

Comments and Suggestions for Authors

The review concentrates on the issue of chronic wounds, the causes of chronicity, and the development of recurrence in an epigenetic context.

Chronic wound pathophysiology goes beyond simple tissue damage and includes complex epigenetic modifications that fundamentally alter cellular behavior and impair healing processes. It is urgent and important to explore the intricate relationship between epigenetic mechanisms and chronic wound healing. The task involves uncovering molecular changes that are responsible for persistent inflammation, tissue repair failure, and the development of therapeutic resistance in chronic wounds.

Chronic wounds are a state of stopped healing that is characterized by persistent inflammation, impaired cell migration, and failure to regenerate tissue. The central mechanisms underlying these pathological processes have been identified as epigenetic modifications by recent advancements in molecular wound biology. The potential for reversibility of epigenetic changes makes them an attractive therapeutic target for promoting wound healing and preventing chronicity, unlike genetic mutations.

The list of references has 60% of articles from the last five years, 15% from 10 years ago, and 8 self-citations. All self-quotes are proper and necessary. The authors have been working in this field for a significant amount of time.

I have some questions and comments for the authors.

  1. The review highlights the epigenetic involvement in the chronicity. However, in the text there is nothing about epigenetic modification generally and in particular regarding chronicity and recurrence of discussed pathologies.
  2. Schemes of pathological mechanisms are present, but ones of epigenetic processes are absent.
  3. There are some links that are misrepresented (ref. 32, 96).

Author Response

Reviewer 2.

Dear Sir/Mme.

We are delighted to express you our sincere gratitude for the constructive and excellent reviewing of our manuscript. We have considered and responded one-by-one of the questions. Please find below our comments.

Thank you very much.

Jorge Berlanga Acosta. Senior Author.

I have some questions and comments for the authors.

  1. The review highlights the epigenetic involvement in the chronicity. However, in the text there is nothing about epigenetic modification generally and in particular regarding chronicity and recurrence of discussed pathologies. This was undertaken. We first reworded chapter V "Chronic wound cells may harbor a chronicity -recurrence epigenetic memory code" and introduced new information with the corresponding references. We also introduced a new block of information “V.2- Epigenetic drivers in wound chronicity" in which the role of the main epigenetic actors in wound chronicity are described, as well as their contemporary validation as predictive biomarkers. 
  2. Schemes of pathological mechanisms are present, but ones of epigenetic processes are absent. This was undertaken and it is represented in the recently created figure 6.
  3. There are some links that are misrepresented (ref. 32, 96). These were revised.

Round 2

Reviewer 2 Report

Comments and Suggestions for Authors

The manuscript was revised by the authors and significantly added with previously missing information in line with the comments.

In the response to comment 2 of report 1, the authors mentioned the recently created Figure 6 with epigenetic processes. However, in the new version of the manuscript, only the old Figure 6 with Diabetic microangiopathy reproduction in non-diabetic rats is present.

It would be greatly appreciated if a summary table with the main epigenetic drivers could be added to the new section 5.2. Epigenetic drivers in wound chronicity.

Author Response

Dear Sir / Mme.

Please find below our reply to your comments. Thank you so much for your excellent criticism to our manuscript. It certainly contributed to improve it.   

Reviewer 1. "It would be greatly appreciated if a summary table with the main epigenetic drivers could be added to the new section 5.2. Epigenetic drivers in wound chronicity."

Response. Table 1 is within the text in section 4 (formerly section 3.3). We deem that in Section 4 it may be more appropriate and in line with the section text. Column 3 summarizes the main and major epigenetic changes described as requested by the Reviewer.

Reviewer 1.  "The authors mentioned the recently created Figure 6 with epigenetic processes. However, in the new version of the manuscript, only the old Figure 6 with Diabetic microangiopathy reproduction in non-diabetic rats is present.”

Response: The first version of the manuscript included only 5 figures. In the new version (attached), a sixth figure was added. However, it was placed after Figure 4. Therefore, the new figure must be Figure 5 and then, the old Figure 5 became Figure 6. In the letter to the editor, we have mistakenly referred to this new figure as Figure 6, but it is included in the manuscript as Figure 5. As correctly indicated by the Reviewer - The diabetic microangiopathy reproduction in non-diabetic rats is now Figure 6.